# 5-Chloroisoxazoles: A Versatile Starting Material for the Preparation of Amides, Anhydrides, Esters, and Thioesters of 2*H*-Azirine-2-carboxylic Acids

**DOI:** 10.3390/molecules28010275

**Published:** 2022-12-29

**Authors:** Anastasiya V. Agafonova, Mikhail S. Novikov, Alexander F. Khlebnikov

**Affiliations:** Institute of Chemistry, St. Petersburg State University, 7/9 Universitetskaya Naberezhnaya, St. Petersburg 199034, Russia

**Keywords:** isoxazoles, azirines, acylation, amides, esters

## Abstract

Amides, anhydrides, esters, and thioesters of 2*H*-azirine-2-carboxylic acids were prepared by a rapid procedure at room temperature involving FeCl_2_-catalyzed isomerization of 5-chloroisoxazoles to 2*H*-azirine-2-carbonyl chlorides, followed by reaction with N-, O-, or S-nucleophiles mediated by an *ortho*-substituted pyridine. With readily available chloroisoxazoles and a nucleophile, 2-picoline can be used as an inexpensive base. When a high yield of the acylation product is important, the reagent 2-(trimethylsilyl)pyridine/ethyl chloroformate is more suitable for the acylation with 2*H*-azirine-2-carbonyl chlorides.

## 1. Introduction

2*H*-Azirine-2-carboxylic acid derivatives are not only useful synthetic blocks but also exhibit various biological activities [1,2,3]. Accordingly, 2*H*-azirine-2-carboxylic acid esters show antibacterial properties [4,5,6,7,8]. Recently, 2*H*-azirine-2-carboxylates, azirine-containing depsipeptides, 2*H*-azirine-2-carboxamides, and azirine-containing dipeptides, which showed activity against ESKAPE pathogens, were prepared using the Passerini and Ugi reactions. [9]. In addition to the Ugi reaction (Figure 1, Equation (1)), 2*H*-azirine-2-carboxamides were prepared by thermal [10,11], metal-catalyzed [12,13,14,15], and photo-isomerization [11,16,17] of 5-amino-substituted isoxazoles (Figure 1, Equation (2)). A limitation of the isomerization approach is that the required 5-amino-substituted isoxazoles can only be prepared by the reaction of 5-halo-substituted isoxazoles with highly nucleophilic amines, while non-nucleophilic amines, in particular anilines, do not react [18]. In addition, UV irradiation may be incompatible with UV-sensitive substituents, while the use of visible light requires an expensive Hoveyda-Grubbs II (HG-II) catalyst as a photocatalyst [17].

2*H*-Azirine-2-carbonyl chloride **1**, prepared by Fe(II)-catalyzed isomerization of 5-chloroisoxazoles **2** (for the mechanism of isomerization, see [12]), react with some N-nucleophiles [12,19,20,21] to give the products of the nucleophilic acyl substitution reaction. Thus, 2*H*-azirine-2-carbonyl chlorides react with pyrazoles to give 2-(1*H*-pyrazol-1-ylcarbonyl)-2*H*-azirines **3** [12] (Figure 1, Equation (3)), benzotriazole to give 1-(2*H*-azirine-2-carbonyl)benzotriazoles **4** [19], and sodium azide to give 2*H*-azirine-2-carbonyl azides **5** [20,21] (Figure 1, Equation (4)). Surprisingly, the treatment of 2*H*-azirine-2-carbonyl chloride **1a** with 3 equiv. of morpholine **6a** gives no amide **7a** (Figure 1, Equation (5)). Taking into account the limited availability of 2*H*-azirine-2-carboxamides noted above, we set ourselves the task of finding optimal conditions for the acylation of primary and secondary amines with 2*H*-azirine-2-carbonyl chlorides. Here we report the preparation of 2*H*-azirine-2-carboxamides from various amines and 2*H*-azirine-2-carbonyl chlorides, as well as the conversion of the latter into anhydride, esters, and thioesters of 2*H*-azirine-2-carboxylic/thiocarboxylic acids by reactions with O- and S-nucleophiles.

## 2. Results and Discussion

Since the reaction of 2*H*-azirine-2-carbonyl chloride **1a** with 3 equiv. of morpholine, **6a**, gives within 3 min a complex mixture of products, without any amide **7a** among them, the amount of used morpholine was varied. It resulted in the presence of 2 equiv. of morpholine, amide **7a** was formed at 70% yield. Moreover, it was found that when 1 equiv. of morpholine was added to the amide **7a**, its complete destruction occurred within 10 min. This means that the absence of amide **7a** among products of the reaction of 2*H*-azirine-2-carbonyl chloride **1a** with 3 equiv. of morpholine **6a** is due to the destruction of the azirine core of amide **7a** and possibly the azirine core of the starting chloride **1a**. Then, the effect of adding other bases to trap HCl on the yield of amide **7a** was investigated (Table 1). The best yield of amide **7a** (89–90%) was obtained with 2,6-lutidine and 2-picoline (Table 1, entries 4, 5). The use of pyridines without an *ortho*-substituent gave a significantly lower yield of amide **7a** (Table 1, entries 2, 6). Tertiary amines also turned out to be less effective in the reaction (Table 1, entries 7, 8).

We also tested some other approaches to amide **7a**. Thus, heating 5-chloroisoxazole **2a** with morpholine **6a** to form 5-morpholinoisoxazole **8a**, followed by its isomerization catalyzed by Fe(II), gave amide **7a** only with an isolated yield of 60% (Figure 2). Hydrolysis of carbonyl chloride **1a** to azirine-2-carboxylic acid **9a**, followed by amide coupling using HATU/DIPEA reagents, gave amide **7a** in a 77% yield (Figure 2). These results indicated that using 1 equiv. of morpholine in a reaction with chloride **1a** in the presence of 1 equiv. of the *ortho*-substituted pyridine is the most efficient approach in converting chloroisoxazole **2a** to amide **7a**.

Therefore, using the cheapest 2-picoline, a number of cyclic (**6b**,**c**), secondary (**6d**,**e**), and primary (**6f–h**) amines were reacted with chloride **1a** (Figure 3) to give amides **7b–h**.

It resulted that the yields of amides **7b–h** obtained from amines **6b–h** were significantly lower than the yield of morpholide **7a**. This may be due to the complex effect of amine nucleophilicity on the competition of amine attack on the carbonyl of the acid chloride group and on the C=N azirine bond, which affects the ratio of the processes of amide formation and azirine ring destruction. In addition, the difference in the basicity of amines affected the formation and ratio of hydrochlorides of amine **6** and 2-picoline, which complicated the protonation of the azirine nitrogen of compounds **1** and **7**. In turn, this catalyzed the reaction of the amine with the azirine core, and this catalysis accelerated the decomposition of azirines.

Surprisingly when chloride **1a** was treated with 2-picoline without the subsequent addition of any amine, anhydride **10a** was obtained in a 40% yield (Figure 4). Compound **10a** was obtained as a mixture of two diastereomers (~1:1.1). (*RS,SR*)-isomer crystallized from the mixture, and its amazing π-stacked structure was confirmed by X-ray analysis (Figure 1) (See the Appendix A). It was found that chloride **1a** does not react with acid **9a** to give anhydride **10a** in the absence of 2-picoline. A plausible mechanism of the formation of anhydride **10** includes the reaction of salt **11** with acid **9** promoted by 2-picoline. Acid **9** was formed by the hydrolysis of chloride **1** with water adsorbed on the wall of the glassware (the glassware was not specially dried) and traces of water entering the reaction mixture during the isolation of chloride **1** from the mixture obtained after the isomerization of chloroisoxazole **2** (extraction with ether, filtration through celite). Since the scale of the reaction of chloroisoxazole **2a** was 1 mmol, only 3.6 mg of water was required to obtain a 40% yield of anhydride **10a**. When chloroisoxazole **2b** was used in the reaction, anhydride **10b** was obtained at 66% (Figure 4). The addition of 0.2 equiv. of water, together with 2-picoline, increased the yield of anhydride **10b** to 78%. Both the excess and deficiency of water in the reaction mixture will reduce the yield of anhydride, but it is difficult to control the exact amount of water required for the reaction on a 1 mmol scale without technically complicating the reaction procedure.

The formation of anhydride **10** due to the presence of traces water suggests an additional route for the formation of amides through the reaction of an amine with an anhydride. Thus, the formation of amides **7** can proceed via several mechanisms that can operate simultaneously: (1) direct interaction of chloride **1a** with amines **6** (Figure 5, route a); (2) pathways mediated by a pyridine via salt **A** (Figure 5, route b); and (3) formation of anhydride **10**, initiated by traces of water, followed by reaction of this anhydride with amines **6** (Figure 5, route c).

Amines **6** differ considerably in nucleophilicity [22] and basicity [23], which affects the rate of their acylation with azirinecarbonyl chlorides **1** and the relative rate of decomposition of the starting azirines **1** and products **7** (vide supra). For instance, the difference in the relative rate of decomposition of azirine **7h** in the presence of different amines was demonstrated by ^1^H NMR analysis of equimolar mixtures of azirine **7h** with *tert*-butylamine and azirine **7h** with morpholine in C_6_D_6_. In the first mixture, primarily starting substances were detected after 1 h, and there was 85% decomposition of azirine **7h** with the formation of a complex mixture of products after 10 h, while in the second mixture, complete decomposition of azirine **7h** was recorded after 1 h.

In order to partially smooth out the differences in the nucleophilicity and basicity of the amines, we decided to increase the excess of amine to 3 equiv. to accelerate the rate of amide formation and reduce the effect of this excess on the decomposition of azirines and add ethyl chloroformate. We assumed that such an additive could, on the one hand, create an additional pathway for the formation of amide **7** and, on the other hand, neutralize the unfavorable effect of an excess of amine by converting it into amide **E** and a salt of weak acid **F**. The latter, in contrast to the corresponding hydrochloride salt, should not protonate azirines and thus initiate their decomposition by amines (Figure 5, route d). To check this hypothesis, additional experiments were carried out to prepare amide **7a** (Table 2). The best results in terms of the yield of amide **7a** and the economical use of reagents correspond to experiment 3. It was also seen that the use of acetyl chloride (Table 2, entry 6) for the above purposes was much less efficient than ethyl chloroformate, which may be due to the fact that salt **F**, with AcO^-^ instead of EtO_2_CO^-^, can better catalyze the decomposition of azirines, being a salt of a stronger acid.

High yields of amides **7** using pyridines with bulky *ortho*-substituents (Table 2) may be because, on the one hand, the C(O)–N bond in salt **A** (Figure 5) in such compounds is weaker, making the respective pyridinium a better leaving group, and on the other hand, the pyridinium chloride with such *ortho*-substituents has a lower protonation capacity for azirines.

Then, experiments were carried out with various amines, **6a**–**h**, under optimal conditions (entry 3 of Table 2). The use of these reaction conditions allowed for the preparation of amides **7** in higher yields (excluding **7c**): **7a** (90%), **7b** (62%), **7c** (35%), **7d** (75%), **7e** (53%), **7f** (86%), **7g** (72%), and **7h** (91%) (Figure 3, footnote *a*). A dramatic increase in yield was significant in the case of *tert*-butylamine (**6h**). We noted that the yield of amide **7h** was only 42% in the direct reaction of chloride **1a** with 2 equiv. of **6h**. Importantly, the reaction with aniline gave anilide **7f**, which was not available by other approaches [10,11,12,13,14,15,16,17], in an 86% yield.

The scope of the reaction with various 2*H*-azirine-2-carbonyl chlorides **1** was evaluated using *tert*-butyl amine (**6h**) as the N-nucleophile. Amides **7h–q** were obtained in 60–91% yields (Figure 6), starting from 3-aryl/hetaryl/alkyl-2*H*-azirine-2-carbonyl chlorides **1a–i**, as well from 2,2,3-trisubstituted derivative **1j**. All new compounds were characterized by ^1^H, ^13^C NMR, and HRMS methods. Moreover, the structure of **7n** was also confirmed by single-crystal X-ray diffraction analysis (Figure 2) (See the Appendix A).

Additionally, 2*H*-azirine-2-carbonyl chloride **1a** was reacted with alkyl- and aryl-substituted O- and S-nucleophiles, such as phenol **12a**, methyl 2-(2-hydroxyphenyl)acetate **12b**, propargyl **12c**, and benzyl **12d** alcohols, as well as thiophenol **13a**, cyclopentanethiol **13b**, and *N*-(sulfanylmethyl)benzamide **13c**. The reactions were carried out under standard conditions but used 1.5 equiv. of O- or S-nucleophile and with the DMAP addition to trap HCl. Azirinecarboxylic esters **14a-d** were obtained in 77-92% isolated yields (Figure 7). *S*-Phenyl and *S*-cyclopentyl azirinecarbothioates **15a,b** were obtained in 81% and 73% yields; the yield of *S*-(benzamidomethyl) azirinecarbothioate **15c** was lower, only 21%, likely due to multifunctionality of the starting *N*-(sulfanylmethyl)benzamide (Figure 7). The yields of **14a**/**15a** were significantly lower when using 2 equiv. of 2-picoline and 1 equiv. of nucleophile **12a**/**13a** (cf. Table 2).

## 3. Materials and Methods

### 3.1. General Instrumentation

Melting points were determined on a melting point apparatus SMP30. ^1^H-NMR (400 MHz) and ^13^C-NMR (100 MHz) spectra were recorded on a Bruker AVANCE 400 spectrometer in CDCl_3_ or C_6_D_6_. Chemical shifts (δ) were reported in parts per million downfield from tetramethylsilane (TMS, δ = 0.00). ^1^H-NMR spectra were calibrated according to the residual peak of CDCl_3_ (7.28 ppm) and C_6_D_6_ (7.16 ppm). For all new compounds, ^13^C{^1^H} spectra were recorded and calibrated according to the peak of CDCl_3_ (77.00 ppm) and C_6_D_6_ (128.00 ppm). Electrospray ionization (ESI), positive mode, and mass spectra were measured on a Bruker MaXis mass spectrometer, HRMS-ESI-QTOF, using MeOH for the dilution of samples. Single-crystal X-ray data were collected by means of “XtaLAB Synergy” and “SuperNova” diffractometers. The crystals of **7n** and **10a** were measured at a temperature of 99.9(8) K. Crystallographic data for the structures **7n** (CCDC 2214535) and **10a** (CCDC 2215155) have been deposited with the Cambridge Crystallographic Data Centre. Thin-layer chromatography (TLC) was conducted on aluminum sheets precoated with SiO_2_ ALUGRAM SIL G/UV254. Column chromatography was performed on Macherey-Nagel silica gel 60M (0.04–0.063 mm). All solvents were distilled and dried prior to use. Toluene and diethyl ether were distilled and stored over sodium metal. MeCN was distilled from P_2_O_5_ and redistilled from K_2_CO_3_. 5-Chloroisoxazoles **2a** [18], **2b** [24], **2f,g,i** [19], **2h** [20], and **2j** [25] are known compounds and were prepared by the reported procedures.

### 3.2. General Experimental Procedures

#### 3.2.1. General Procedure A (GP-A) for the Synthesis of 5-Chloroisoxazoles (**2**)

Triethylamine (253 mg, 2.5 mmol, 0.35 mL) was added dropwise at 0 °C to a stirring suspension of isoxazol-5(4*H*)-one (3 mmol) in POCl_3_ (4 mL). The mixture was stirred at 75 °C for 12 h, poured into ice (300 g), and extracted with EtOAc (3 × 30 mL). The organic layer was washed with brine and dried over Na_2_SO_4_. The solvent was evaporated under reduced pressure, and the product was purified by column chromatography (petroleum ether–EtOAc, 10:1) to give chloroisoxazole **2**.

#### 3.2.2. General Procedure B (GP-B) for the Synthesis of Amides (**7**)

Anhydrous FeCl_2_ (51 mg, 0.4 mmol, 0.2 equiv.) was added to a solution of 5-chloroisoxazole **2** (2 mmol) in dry acetonitrile (4 mL) under an Ar atmosphere. The mixture was stirred at room temperature for 2 h until 5-chloroisoxazole **2** was consumed (monitored by TLC). The solvent was evaporated, the residue was diluted with dry diethyl ether (100 mL), and the precipitated iron chloride was filtered off through celite. The ether was evaporated under reduced pressure, and the formed 2*H*-azirine-2-carbonyl chloride **1** was dissolved in anhydrous toluene (4 mL). Then amine **6** (4 mmol, 2 equiv.) was added, and the reaction mixture was stirred at room temperature for 3 min. The solvent was evaporated, and the residue was purified by column chromatography (petroleum ether–EtOAc).

#### 3.2.3. General Procedure C (GP-C) for the Synthesis of Amides (**7**)

Anhydrous FeCl_2_ (51 mg, 0.4 mmol, 0.2 equiv.) was added to a solution of 5-chloroisoxazole **2** (2 mmol) in dry acetonitrile (4 mL) under an Ar atmosphere. The mixture was stirred at room temperature for 2 h until 5-chloroisoxazole **2** was consumed (monitored by TLC). The solvent was evaporated, the residue was diluted with dry diethyl ether (100 mL), and the precipitated iron chloride was filtered off through celite. The ether was evaporated under reduced pressure, and the formed 2*H*-azirine-2-carbonyl chloride **1** was dissolved in anhydrous toluene (4 mL). Then 2-methylpyridine (186 mg, 2 mmol, 1 equiv.) and amine **6** (2 mmol, 1 equiv.) were added successively. The reaction mixture was stirred for 10 min at room temperature, the solvent was evaporated, and the residue was purified by column chromatography (petroleum ether–EtOAc).

#### 3.2.4. General Procedure D (GP-D) for the Synthesis of Amides (**7**)

Anhydrous FeCl_2_ (51 mg, 0.4 mmol, 0.2 equiv.) was added to a solution of 5-chloroisoxazole **2** (2 mmol) in dry acetonitrile (4 mL) under an Ar atmosphere. The mixture was stirred at room temperature for 2 h until 5-chloroisoxazole **2** was consumed (monitored by TLC). The solvent was evaporated, the residue was diluted with dry diethyl ether (100 mL), and the precipitated iron chloride was filtered off through celite. The ether was evaporated under reduced pressure, and the formed 2*H*-azirine-2-carbonyl chloride **1** was dissolved in anhydrous toluene (2 mL). The resulting solution was added to a stirred mixture of 2-(trimethylsilyl)pyridine (151 mg, 1 mmol, 0.5 equiv.) and ethyl chloroformate (109 mg, 1 mmol, 0.5 equiv.) at room temperature. Then amine **6** (6 mmol, 3 equiv.) was added, and the reaction mixture was stirred for 5 min. The solvent was evaporated, and the residue was purified by column chromatography (petroleum ether–EtOAc).

#### 3.2.5. General Procedure E (GP-E) for the Synthesis of Amides (**7**)

A mixture of 5-chloroisoxazole **2** (2 mmol), amine (4 mmol, 2 equiv.), and K_2_CO_3_ (828 mg, 6 mmol, 3 equiv.) in DMF (10 mL) was refluxed while being stirred for 1.5 h. The reaction mixture was diluted with water (30 mL), extracted with EtOAc (3 × 15 mL), washed with brine (20 mL), and dried over Na_2_SO_4_. The solvent was evaporated, the residue was dissolved in acetonitrile (10 mL), and FeCl_2_ × 4H_2_O (80 mg, 20 mol %) was added. The mixture was stirred at room temperature for 3 h, the solvent was evaporated, and the residue was purified by column chromatography (petroleum ether–EtOAc).

#### 3.2.6. General Procedure F (GP-F) for the Synthesis of Amides (**7**)

Anhydrous FeCl_2_ (51 mg, 0.4 mmol, 0.2 equiv.) was added to a solution of 5-chloroisoxazole **2** (2 mmol) in dry acetonitrile (25 mL) under an Ar atmosphere. The mixture was stirred at room temperature for 2 h until 5-chloroisoxazole **2** was consumed (monitored by TLC). Water (25 mL) was added, and the mixture was stirred at room temperature for 15 min. The 2*H*-azirine-2-carboxylic acid was extracted with EtOAc (325 mL), washed with water (25 mL) and brine (10 mL), and dried over Na_2_SO_4_. The solvent was evaporated, and the residue was diluted with DMF (5 mL) and cooled to 0 °C. Then HATU (1.52 g, 4 mmol, 2 equiv.) and DIPEA (774 mg, 6 mmol, 3 equiv.) were added, and the reaction mixture was stirred at 0 °C for 10 min. After that, amine **6** (2.4 mmol, 1.2 equiv) was added, and the reaction mixture was stirred at room temperature for 10 min. The reaction mixture was then diluted with water (75 mL), extracted with EtOAc (3 × 25 mL), washed with brine (30 mL), and dried over Na_2_SO_4_. The solvent was evaporated, and the residue was purified by column chromatography (petroleum ether–EtOAc).

#### 3.2.7. General Procedure G (GP-G) for the Synthesis of Anhydrides (**10**)

Anhydrous FeCl_2_ (51 mg, 0.4 mmol, 0.2 equiv.) was added to a solution of 5-chloroisoxazole **2** (2 mmol) in dry acetonitrile (4 mL) under an Ar atmosphere. The mixture was stirred at room temperature for 2 h until 5-chloroisoxazole **2** was consumed (monitored by TLC). The solvent was evaporated, the residue was diluted with dry diethyl ether (100 mL), and the precipitated iron chloride was filtered off through celite. The ether was evaporated, and the formed 2*H*-azirine-2-carbonyl chloride **1** was dissolved in anhydrous toluene (4 mL). Then 2-methylpyridine (186 mg, 2 mmol, 1 equiv.) was added. The reaction mixture was stirred for 10 min at room temperature, the solvent was evaporated, and the residue was purified by column chromatography (petroleum ether–EtOAc).

#### 3.2.8. General Procedure H (GP-H) for the Synthesis of Esters (**14**) and Thioesters (**15**)

Anhydrous FeCl_2_ (51 mg, 0.4 mmol, 0.2 equiv.) was added to a solution of 5-chloroisoxazole **2a** (359 mg, 2 mmol) in dry acetonitrile (4 mL) under an Ar atmosphere. The mixture was stirred at room temperature for 2 h until 5-chloroisoxazole **2a** was consumed (monitored by TLC). The solvent was evaporated, the residue was diluted with dry diethyl ether (100 mL), and the precipitated iron chloride was filtered off through celite. The ether was evaporated, anhydrous toluene (2 mL) was added, and the resulting solution was added to a stirring mixture of 2-(trimethylsilyl)pyridine (151 mg, 1 mmol, 0.5 equiv.) and ethyl chloroformate (109 mg, 1 mmol, 0.5 equiv) at room temperature. Then the corresponding nucleophile (3 mmol, 1.5 equiv.) and *N*,*N*-dimethylpyridin-4-amine (733 mg, 6 mmol, 3 equiv.) base were added, and the reaction mixture was stirred for 10 min. The solvent was evaporated, and the residue was purified by column chromatography (petroleum ether–EtOAc).

#### 3.2.9. Specific Procedures and Characterization

*3-(Benzo[d]**[1,3]**dioxol-5-yl)-5-chloroisoxazole* (**2c**). Compound **2c** was prepared following the general procedure of GP-A from 3-(benzo[*d*][1,3]dioxol-5-yl)isoxazol-5(4*H*)-one (616 mg, 3 mmol) as colorless solid (443 mg, yield 66%). Mp: 82–83 °C (Et_2_O–hexane). ^1^H NMR (400 MHz, CDCl_3_) *δ* 7.28 (s, 1H), 7.25–7.20 (m, 1H), 6.93–6.85 (m, 1H), 6.41 (s, 1H), 6.05 (s, 2H). ^13^C{1H} NMR (100 MHz, CDCl_3_) *δ* 163.7, 154.9, 149.6, 148.3, 122.1, 121.2, 108.7, 106.6, 101.6, 99.4. HRMS-ESI [M + H]^+^ calcd for C_10_H_7_^35^ClNO_3_^+^, 224.0109; found, 224.0112.

*5-Chloro-3-(2,3-dihydrobenzo[b]**[1,4]**dioxin-6-yl)isoxazole* (**2d**). Compound **2d** was prepared following the general procedure of GP-A from 3-(2,3-dihydrobenzo[*b*][1,4]dioxin-6-yl)isoxazol-5(4*H*)-one (658 mg, 3 mmol) as colorless solid (506 mg, yield 71%). Mp: 78–79 °C (Et_2_O–hexane). ^1^H NMR (400 MHz, CDCl_3_) *δ* 7.29 (s, 1H), 7.26–7.21 (m, 1H), 6.96–6.89 (m, 1H), 6.39 (s, 1H), 4.36–4.25 (s, 4H). ^13^C{1H} NMR (100 MHz, CDCl_3_) *δ* 163.6, 154.7, 145.6, 143.9, 121.4, 120.0, 117.8, 115.6, 99.4, 64.5, 64.2. HRMS-ESI [M + Na]^+^ calcd for C_11_H_8_^35^ClNNaO_3_^+^, 260.0085; found, 260.0079.

*5-Chloro-3-(2,3-dihydrobenzo[b]*[1,4]*dioxin-6-yl)isoxazole* (**2e**). Compound **2e** was prepared following the general procedure of GP-A from 3-([1,1’-biphenyl]-4-yl)isoxazol-5(4*H*)-one (712 mg, 3 mmol) as colorless solid (506 mg, yield 66%). Mp: 139–140 °C (Et_2_O–hexane). ^1^H NMR (400 MHz, CDCl_3_) *δ* 7.89–7.83 (m, 2H), 7.75–7.71 (m, 2H), 7.68–7.63 (m, 2H), 7.53–7.47 (m, 2H), 7.46–7.39 (m, 1H), 6.54 (s, 1H). ^13^C{1H} NMR (100 MHz, CDCl_3_) *δ* 163.9, 155.1, 143.4, 140.0, 128.9, 127.9, 127.7 (2C), 127.1, 127.0, 99.6. HRMS-ESI [M + H]^+^ calcd for C_15_H_11_^35^ClNO^+^, 256.0524; found, 256.0527.

*Morpholino(3-phenyl-2H-azirin-2-yl)methanone* (**7a**). Compound **7a** was prepared following GP-B–F procedure from 5-chloro-3-phenylisoxazole **2a** (359 mg, 2 mmol, 1 equiv.) with morpholine **6a** (GP-B and GP-E: 348 mg, 4 mmol, 2 equiv.; GP-C: 174 mg, 2 mmol, 1 equiv.; GP-D: 522 mg, 6 mmol, 3 equiv.; GP-F: 209 mg, 2.4 mmol, 1.2 equiv.) as a nucleophile. A mixture of PE–EtOAc (from 5:1 to 1:1) was used as an eluent for chromatography. Orange solid (GP-B: 322 mg, yield 70%; GP-C and GP-D: 506 mg, yield 90%; GP-E: 276 mg, yield 60%; GP-F: 355 mg, yield 77%). Mp: 73–75 °C (Et_2_O–hexane). ^1^H NMR (400 MHz, CDCl_3_) *δ* 7.96–7.82 (m, 2H), 7.65–7.51 (m, 3H), 3.97–3.57 (m, 8H), 3.04 (s, 1H). ^13^C{1H} NMR (100 MHz, CDCl_3_) *δ* 169.0, 159.3, 133.5, 130.2, 129.1, 122.9, 66.7 (2C), 45.9, 42.6, 28.5. HRMS-ESI [M + Na]^+^ calcd for C_13_H_14_N_2_NaO_2_^+^, 253.0948; found, 253.0949.

*(4-Methylpiperidin-1-yl)(3-phenyl-2H-azirin-2-yl)methanone* (**7b**). Compound **7b** was prepared following GP-C and GP-D procedures from 5-chloro-3-phenylisoxazole **2a** (359 mg, 2 mmol, 1 equiv) with 4-methylpiperidine **6b** (GP-C: 198 mg, 2 mmol, 1 equiv; GP-D: 595 mg, 6 mmol, 3 equiv) as a nucleophile. A mixture of PE–EtOAc (from 5:1 to 1:1) was used as an eluent for chromatography. Orange oil (GP-C: 325 mg, yield 67%; GP-D: 300 mg, yield 62%). ^1^H NMR (400 MHz, CDCl_3_) *δ* 7.99–7.86 (m, 2H), 7.66–7.52 (m, 3H), 4.64–4.39 (m, 2H), 3.38–3.20 (m, 1H), 3.10 (s, 1H), 2.78–2.60 (m, 1H), 1.95–1.79 (m, 1H), 1.77–1.65 (m, 2H), 1.41–1.13 (m, 2H), 1.02 (d, *J* = 6.2 Hz, 3H). ^13^C{1H} NMR (100 MHz, CDCl_3_) *δ* 168.5, 159.8 (d, *J* = 17.5 Hz), 133.3, 130.2, 129.1, 123.3, 45.9 (d, *J* = 11.6 Hz), 42.8 (d, *J* = 13.9 Hz), 34.9 (d, *J* = 13.3 Hz), 33.6, 31.1 (d, *J* = 13.1 Hz), 29.0, 21.7. HRMS-ESI [M + H]^+^ calcd for C_15_H_19_N_2_O^+^, 243.1492; found, 243.1494.

*(3-Phenyl-2H-azirin-2-yl)(pyrrolidin-1-yl)methanone* (**7c**). Compound **7c** was prepared following GP-C and GP-D procedures from 5-chloro-3-phenylisoxazole **2a** (359 mg, 2 mmol, 1 equiv.) with pyrrolidine **6c** (GP-C: 142 mg, 2 mmol, 1 equiv.; GP-D: 427 mg, 6 mmol, 3 equiv.) as a nucleophile. A mixture of PE–EtOAc (from 5:1 to 1:1) was used as an eluent for chromatography. Orange solid (GP-C: 309 mg, yield 72%; GP-D: 150 mg, yield 35%). Mp: 58–59 °C (Et_2_O–hexane). ^1^H NMR (400 MHz, CDCl_3_) *δ* 7.94–7.84 (m, 2H), 7.64–7.50 (m, 3H), 3.93–3.74 (m, 2H), 3.61–3.49 (m, 2H), 2.97 (s, 1H), 2.13–1.83 (m, 4H). ^13^C{1H} NMR (100 MHz, CDCl_3_) *δ* 168.6, 159.4, 133.3, 130.2, 129.0, 123.1, 46.4, 46.3, 30.0, 26.2, 24.1. HRMS-ESI [M + H]^+^ calcd for C_13_H_15_N_2_O^+^, 215.1179; found, 215.1181.

***N****,N-Diethyl-3-phenyl-2H-azirine-2-carboxamide* (**7d**). Compound **7d** was prepared following GP-C and GP-D procedures from 5-chloro-3-phenylisoxazole **2a** (359 mg, 2 mmol, 1 equiv.) with diethylamine **6d** (GP-C: 146 mg, 2 mmol, 1 equiv.; GP-D: 439 mg, 6 mmol, 3 equiv.) as a nucleophile. A mixture of PE–EtOAc (from 5:1 to 1:1) was used as an eluent for chromatography. Orange solid (GP-C: 260 mg, yield 60%; GP-D: 324 mg, yield 75%). Mp: 53–54 °C (Et_2_O–hexane). ^1^H NMR (400 MHz, CDCl_3_) *δ* 7.92–7.86 (m, 2H), 7.62–7.51 (m, 3H), 3.80–3.70 (m, 1H), 3.70–3.59 (m, 1H), 3.58–3.48 (m, 1H), 3.44–3.34 (m, 1H), 3.04 (s, 1H), 1.40 (t, *J* = 7.1 Hz, 3H), 1.16 (t, *J* = 7.1 Hz, 3H). ^13^C{1H} NMR (100 MHz, CDCl_3_) *δ* 169.4, 159.4, 133.2, 130.1, 129.0, 123.3, 42.0, 41.1, 29.0, 15.1, 13.0. HRMS-ESI [M + H]^+^ calcd for C_13_H_17_N_2_O^+^, 217.1335; found, 217.1338.

*N,N,3-Triphenyl-2H-azirine-2-carboxamide* (**7e**). Compound **7e** was prepared following GP-C and GP-D procedures from 5-chloro-3-phenylisoxazole **2a** (359 mg, 2 mmol, 1 equiv.) with diphenylamine **6e** (GP-C: 334 mg, 2 mmol, 1 equiv.; GP-D: 1.02 g, 6 mmol, 3 equiv.) as a nucleophile. A mixture of PE–EtOAc (from 5:1 to 1:1) was used as an eluent for chromatography. Orange solid (GP-C: 231 mg, yield 37%; GP-D: 331 mg, yield 53%). Mp: 189–190 °C (Et_2_O–hexane). ^1^H NMR (400 MHz, CDCl_3_) *δ* 7.99–7.87 (m, 2H), 7.67–7.56 (m, 3H), 7.53–7.24 (m, 10H), 2.82 (s, 1H). ^13^C{1H} NMR (100 MHz, CDCl_3_) *δ* 170.4, 158.6, 142.4, 133.4, 130.2, 129.4, 129.1, 127.1, 123.0, 31.3. HRMS-ESI [M + H]^+^ calcd for C_21_H_17_N_2_O^+^, 313.1335; found, 313.1340.

*N,3-Diphenyl-2H-azirine-2-carboxamide* (**7f**). Compound **7f** was prepared following GP-C and GP-D procedures from 5-chloro-3-phenylisoxazole **2a** (359 mg, 2 mmol, 1 equiv.) with aniline **6f** (GP-C: 186 mg, 2 mmol, 1 equiv; GP-D: 559 mg, 6 mmol, 3 equiv.) as a nucleophile. A mixture of PE–EtOAc (from 5:1 to 1:1) was used as an eluent for chromatography. Orange solid (GP-C: 350 mg, yield 74%; GP-D: 406 mg, yield 86%). Mp: 179–180 °C (Et_2_O–hexane). ^1^H NMR (400 MHz, CDCl_3_) *δ* 8.01–7.93 (m, 2H), 7.71–7.65 (m, 1H), 7.63–7.57 (m, 2H), 7.55–7.50 (m, 2H), 7.43 (s, 1H), 7.34–7.26 (m, 2H), 7.13–7.07 (m, 1H), 2.94 (s, 1H). ^13^C{1H} NMR (100 MHz, CDCl_3_) *δ* 168.6, 162.2, 137.4, 134.3, 130.6, 129.4, 128.9, 124.4, 122.3, 119.7, 32.0. HRMS-ESI [M + Na]^+^ calcd for C_15_H_12_N_2_NaO^+^, 259.0842; found, 259.0842.

*N-Benzyl-3-phenyl-2H-azirine-2-carboxamide* (**7g**) [16,17]. Compound **7g** was prepared following GP-C and GP-D procedures from 5-chloro-3-phenylisoxazole **2a** (359 mg, 2 mmol, 1 equiv.) with phenylmethanamine **6g** (GP-C: 214 mg, 2 mmol, 1 equiv.; GP-D: 643 mg, 6 mmol, 3 equiv.) as a nucleophile. A mixture of PE–EtOAc (from 5:1 to 1:1) was used as an eluent for chromatography. Light yellow solid (GP-C: 280 mg, yield 56%; GP-D: 360 mg, yield 72%). Mp: 145–146 °C (Et_2_O–hexane). ^1^H NMR (400 MHz, CDCl_3_) *δ* 7.96–7.89 (m, 2H), 7.70–7.65 (m, 1H), 7.63–7.57 (m, 2H), 7.33–7.20 (m, 5H), 5.90 (s, 1H), 4.54–4.35 (m, 2H), 2.85 (s, 1H). ^13^C{1H} NMR (100 MHz, CDCl_3_) *δ* 170.5, 162.1, 137.9, 134.1, 130.5, 129.4, 128.6, 127.6, 127.4, 122.4, 43.4, 31.4. HRMS-ESI [M + Na]^+^ calcd for C_16_H_14_N_2_NaO^+^, 273.0998; found, 273.1001.

*N-(tert-Butyl)-3-phenyl-2H-azirine-2-carboxamide* (**7h**). Compound **7h** was prepared following methods GP-B, GP-C and GP-D from 5-chloro-3-phenylisoxazole **2a** (359 mg, 2 mmol, 1 equiv.) with 2-methylpropan-2-amine **6h** (GP-B: 293 mg, 4 mmol, 2 equiv.; GP-C: 146 mg, 2 mmol, 1 equiv.; GP-D: 439 mg, 6 mmol, 3 equiv.) as a nucleophile. A mixture of PE–EtOAc (from 5:1 to 1:1) was used as an eluent for chromatography. Colorless solid (GP-B: 182 mg, yield 42%; GP-C: 134 mg, yield 31%; GP-D: 394 mg, yield 91%). Mp: 171–172 °C (Et_2_O–hexane). ^1^H NMR (400 MHz, CDCl_3_) *δ* 7.95–7.89 (m, 2H), 7.69–7.63 (m, 1H), 7.63–7.56 (m, 2H), 5.32 (s, 1H), 2.69 (s, 1H), 1.32 (s, 9H). ^13^C{1H} NMR (100 MHz, CDCl_3_) *δ* 169.6, 162.6, 133.9, 130.3, 129.3, 122.7, 51.3, 32.1, 28.7. HRMS-ESI [M + H]^+^ calcd for C_13_H_17_N_2_O^+^, 217.1335; found, 217.1338.

*N-(tert-Butyl)-3-(4-methoxyphenyl)-2H-azirine-2-carboxamide* (**7i**). Compound **7i** was prepared following GP-D procedure from 5-chloro-3-(4-methoxyphenyl)isoxazole **2b** (419 mg, 2 mmol, 1 equiv.) with 2-methylpropan-2-amine **6h** (439 mg, 6 mmol, 3 equiv.) as a nucleophile. A mixture of PE–EtOAc (from 5:1 to 1:1) was used as an eluent for chromatography. Colorless solid (394 mg, yield 80%). Mp: 157–158 °C (Et_2_O–hexane). ^1^H NMR (400 MHz, CDCl_3_) *δ* 7.88–7.82 (m, 2H), 7.10–7.04 (m, 2H), 5.26 (s, 1H), 3.91 (s, 3H), 2.63 (s, 1H), 1.30 (s, 9H). ^13^C{1H} NMR (100 MHz, CDCl_3_) *δ* 170.0, 164.1, 161.3, 132.5, 115.0, 114.9, 55.6, 51.1, 31.8, 28.6. HRMS-ESI [M + Na]^+^ calcd for C_14_H_18_N_2_NaO_2_^+^, 269.1260; found, 269.1262.

*3-(Benzo[d]**[1,3]**dioxol-5-yl)-N-(tert-butyl)-2H-azirine-2-carboxamide* (**7j**). Compound **7j** was prepared following GP-D procedure from 3-(benzo[*d*][1,3]dioxol-5-yl)-5-chloroisoxazole **2c** (447 mg, 2 mmol, 1 equiv.) with 2-methylpropan-2-amine **6h** (439 mg, 6 mmol, 3 equiv.) as a nucleophile. A mixture of PE–EtOAc (from 5:1 to 1:1) was used as an eluent for chromatography. Light yellow solid (422 mg, yield 81%). Mp: 150–151 °C (Et_2_O–hexane). ^1^H NMR (400 MHz, CDCl_3_) *δ* 7.45–7.41 (m, 1H), 7.38–7.35 (m, 1H), 7.01–6.96 (m, 1H), 6.11 (s, 2H), 5.27 (s, 1H), 2.63 (s, 1H), 1.31 (s, 9H). ^13^C{1H} NMR (100 MHz, CDCl_3_) *δ* 169.7, 161.6, 152.6, 148.6, 127.2, 116.5, 109.1, 109.0, 102.2, 51.2, 32.3, 28.6. HRMS-ESI [M + H]^+^ calcd for C_14_H_17_N_2_O_3_^+^, 261.1234; found, 261.1237.

*N-(tert-Butyl)-3-(2,3-dihydrobenzo[b]**[1,4]**dioxin-6-yl)-2H-azirine-2-carboxamide* (**7k**). Compound **7k** was prepared following GP-D procedure from 5-chloro-3-(2,3-dihydrobenzo[*b*][1,4]dioxin-6-yl)isoxazole **2d** (475 mg, 2 mmol, 1 equiv.) with 2-methylpropan-2-amine **6h** (439 mg, 6 mmol, 3 equiv.) as a nucleophile. A mixture of PE–EtOAc (from 5:1 to 1:1) was used as an eluent for chromatography. Colorless solid (455 mg, yield 83%). Mp: 123–124 °C (Et_2_O–hexane). ^1^H NMR (400 MHz, CDCl_3_) *δ* 7.46–7.39 (m, 2H), 7.08–7.00 (m, 1H), 5.24 (s, 1H), 4.38–4.29 (m, 4H), 2.61 (s, 1H), 1.31 (s, 9H). ^13^C{1H} NMR (100 MHz, CDCl_3_) *δ* 169.9, 161.5, 148.7, 144.1, 124.4, 119.3, 118.4, 115.7, 64.7, 64.0, 51.2, 32.0, 28.6. HRMS-ESI [M + Na]^+^ calcd for C_15_H_18_N_2_NaO_3_^+^, 297.1210; found, 297.1212.

*3-([1,1’-Biphenyl]-4-yl)-N-(tert-butyl)-2H-azirine-2-carboxamide* (**7l**). Compound **7l** was prepared following GP-D procedure from 3-([1,1’-biphenyl]-4-yl)-5-chloroisoxazole **2e** (511 mg, 2 mmol, 1 equiv.) with 2-methylpropan-2-amine **6h** (439 mg, 6 mmol, 3 equiv.) as a nucleophile. A mixture of PE–EtOAc (from 5:1 to 1:1) was used as an eluent for chromatography. Light yellow solid (421 mg, yield 72%). Mp: 161–162 °C (Et_2_O–hexane). ^1^H NMR (400 MHz, CDCl_3_) *δ* 8.02–7.98 (m, 2H), 7.84–7.80 (m, 2H), 7.68–7.64 (m, 2H), 7.54–7.42 (m, 3H), 5.33 (s, 1H), 2.72 (s, 1H), 1.35 (s, 9H). ^13^C{1H} NMR (100 MHz, CDCl_3_) *δ* 169.7, 162.3, 146.8, 139.5, 130.8, 129.0, 128.6, 128.0, 127.3, 121.3, 51.3, 32.1, 28.7. HRMS-ESI [M + H]^+^ calcd for C_19_H_21_N_2_O^+^, 293.1648; found, 293.1652.

*3-(4-Bromophenyl)-N-(tert-butyl)-2H-azirine-2-carboxamide* (**7m**). Compound **7m** was prepared following GP-D procedure from 3-(4-bromophenyl)-5-chloroisoxazole **2f** (517 mg, 2 mmol, 1 equiv.) with 2-methylpropan-2-amine **6h** (439 mg, 6 mmol, 3 equiv.) as a nucleophile. A mixture of PE–EtOAc (from 5:1 to 1:1) was used as an eluent for chromatography. Colorless solid (449 mg, yield 76%). Mp: 171–172 °C (Et_2_O–hexane). ^1^H NMR (400 MHz, CDCl_3_) *δ* 7.81–7.72 (m, 4H), 5.34 (s, 1H), 2.69 (s, 1H), 1.33 (s, 9H). ^13^C{1H} NMR (100 MHz, CDCl_3_) *δ* 169.2, 162.1, 132.8, 131.5, 129.1, 121.6, 51.4, 32.3, 28.7. HRMS-ESI [M + H]^+^ calcd for C_13_H_16_^79^BrN_2_O^+^, 295.0441; found, 295.0443.

*N-(tert-butyl)-3-(4-(trifluoromethyl)phenyl)-2H-azirine-2-carboxamide* (**7n**). Compound **7n** was prepared following GP-D procedure from 5-chloro-3-(4-(trifluoromethyl)phenyl)isoxazole **2g** (495 mg, 2 mmol, 1 equiv.) with 2-methylpropan-2-amine **6h** (439 mg, 6 mmol, 3 equiv.) as a nucleophile. A mixture of PE–EtOAc (from 5:1 to 1:1) was used as an eluent for chromatography. Colorless solid (341 mg, yield 60%). Mp: 142–143 °C (Et_2_O–hexane). ^1^H NMR (400 MHz, CDCl_3_) *δ* 8.07–8.01 (m, 2H), 7.88–7.83 (m, 2H), 5.47 (s, 1H), 2.75 (s, 1H), 1.34 (s, 9H). ^13^C{1H} NMR (100 MHz, CDCl_3_) *δ* 168.9, 162.2, 135.3 (q, *J* = 33.0 Hz), 130.6, 126.4 (q, *J* = 3.7 Hz), 126.2, 123.3 (q, *J* = 272.9 Hz), 51.6, 32.6, 28.7. HRMS-ESI [M + H]^+^ calcd for C_14_H_16_F_3_N_2_O^+^, 285.1209; found, 285.1213.

*N-(tert-butyl)-3-(thiophen-2-yl)-2H-azirine-2-carboxamide* (**7o**). Compound **7o** was prepared following GP-D procedure from 5-chloro-3-(thiophen-2-yl)isoxazole **2h** (371 mg, 2 mmol, 1 equiv.) with 2-methylpropan-2-amine **6h** (439 mg, 6 mmol, 3 equiv.) as a nucleophile. A mixture of PE–EtOAc (from 5:1 to 1:1) was used as an eluent for chromatography. Colorless solid (351 mg, yield 79%). Mp: 135–136 °C (Et_2_O–hexane). ^1^H NMR (400 MHz, CDCl_3_) *δ* 7.89 (d, *J* = 3.9 Hz, 1H), 7.74 (d, *J* = 2.7 Hz, 1H), 7.28 (dd, *J* = 5.0, 3.7 Hz, 1H), 5.35 (s, 1H), 2.70 (s, 1H), 1.32 (s, 9H). ^13^C{1H} NMR (100 MHz, CDCl_3_) *δ* 169.2, 155.8, 135.6, 135.5, 128.6, 125.1, 51.3, 32.8, 28.6. HRMS-ESI [M + H]^+^ calcd for C_11_H_15_N_2_Os^+^, 223.0900; found, 223.0903.

*N,3-Di-tert-butyl-2H-azirine-2-carboxamide* (**7p**). Compound **7p** was prepared following GP-D procedure from 3-(*tert*-butyl)-5-chloroisoxazole **2i** (319 mg, 2 mmol, 1 equiv.) with 2-methylpropan-2-amine **6h** (439 mg, 6 mmol, 3 equiv.) as a nucleophile. A mixture of PE–EtOAc (from 5:1 to 1:1) was used as an eluent for chromatography. Colorless solid (322 mg, yield 82%). Mp: 129–130 °C (Et_2_O–hexane). ^1^H NMR (400 MHz, CDCl_3_) *δ* 5.15 (s, 1H), 2.34 (s, 1H), 1.31 (s, 9H), 1.30 (s, 9H). ^13^C{1H} NMR (100 MHz, CDCl_3_) *δ* 13C NMR (101 MHz, CDCl3) δ 171.9, 170.0, 51.1, 33.3, 32.3, 28.7, 25.7. HRMS-ESI [M + H]^+^ calcd for C_11_H_21_N_2_O^+^, 197.1648; found, 197.1651.

*N-(tert-Butyl)-2-methyl-3-phenyl-2H-azirine-2-carboxamide* (**7q**). Compound **7q** was prepared following GP-D procedure from 5-chloro-4-methyl-3-phenylisoxazole **2j** (387 mg, 2 mmol, 1 equiv.) with 2-methylpropan-2-amine **6h** (439 mg, 6 mmol, 3 equiv.) as a nucleophile. A mixture of PE–EtOAc (from 10:1 to 5:1) was used as an eluent for chromatography. Colorless solid (419 mg, yield 91%). Mp: 84–85 °C (Et_2_O–hexane).^1^H NMR (400 MHz, CDCl_3_) *δ* 7.91–7.85 (m, 2H), 7.68–7.63 (m, 1H), 7.62–7.56 (m, 2H), 5.22 (s, 1H), 1.60 (s, 3H), 1.27 (s, 9H). ^13^C{1H} NMR (100 MHz, CDCl_3_) *δ* 171.5, 168.0, 133.8, 130.1, 129.4, 122.7, 51.0, 37.1, 28.5, 17.4. HRMS-ESI [M + H]^+^ calcd for C_14_H_19_N_2_O^+^, 231.1492; found, 231.1495.

*3-Phenyl-2H-azirine-2-carboxylic anhydride* (**10a**). Compound **10a** was prepared following GP-G procedure from 5-chloro-3-phenylisoxazole **2a** (359 mg, 2 mmol, 1 equiv.). A mixture of PE–EtOAc (from 10:1 to 5:1) was used as an eluent for chromatography. Colorless solid (122 mg, yield 40%). Mp: 92–93 °C (Et_2_O–hexane). ^1^H NMR (400 MHz, C_6_D_6_) *δ* 7.32–7.23 (m, 4H), 7.03–6.95 (m, 2H), 6.92–6.83 (m, 4H), 2.80–2.74 (m, 2H). ^13^C{1H} NMR (100 MHz, C_6_D_6_) *δ* 174.21, 174.18, 158.0 (2C), 134.2, 131.1, 130.7 (2C), 129.44, 129.41, 121.10, 121.07, 37.7 (2C). HRMS-ESI [M + Na]^+^ calcd for C_18_H_12_N_2_NaO_3_^+^, 327.0740; found, 327.0733.

*3-(4-Methoxyphenyl)-2H-azirine-2-carboxylic anhydride* (**10b**). Compound **10b** was prepared following GP-G procedure from **2b** (419 mg, 2 mmol, 1 equiv.). A mixture of PE–EtOAc (from 10:1 to 2:1) was used as an eluent for chromatography. Colorless solid (242 mg, yield 66%). Mp: 96–98 °C (Et_2_O–hexane). ^1^H NMR (400 MHz, C_6_D_6_) *δ* 7.49–7.37 (m, 4H), 6.53–6.43 (m, 4H), 3.12 (s, 6H), 2.78–2.60 (m, 2H). ^13^C{1H} NMR (100 MHz, CDCl_3_) *δ* 177.3 (2C), 168.1, 167.9, 164.4, 164.3, 155.8, 155.7, 132.82, 132.79, 115.0, 114.9, 113.4, 113.3, 55.62, 55.60, 29.7, 28.7. HRMS-ESI [M + Na]^+^ calcd for C_20_H_16_N_2_NaO_5_^+^, 387.0951; found, 387.0955.

*Phenyl 3-phenyl-2H-azirine-2-carboxylate* (**14a**). Compound **14a** was prepared following GP-H procedure from 5-chloro-3-phenylisoxazole **2a** (359 mg, 2 mmol, 1 equiv.) with phenol **12a** (282 mg, 3 mmol, 1.5 equiv.) as a nucleophile and *N*,*N*-dimethylpyridin-4-amine (733 mg, 6 mmol, 3 equiv.) as a base. A mixture of PE–EtOAc (from 20:1 to 5:1) was used as an eluent for chromatography. Colorless solid (399 mg, yield 84%). Mp: 82–83 °C (Et_2_O–hexane). ^1^H NMR (400 MHz, CDCl_3_) *δ* 8.04–7.95 (m, 2H), 7.74–7.59 (m, 3H), 7.43–7.34 (m, 2H), 7.28–7.21 (m, 1H), 7.18–7.08 (m, 2H), 3.09 (s, 1H). ^13^C{1H} NMR (100 MHz, CDCl_3_) *δ* 170.1, 158.2, 150.6, 134.1, 130.5, 129.4, 129.3, 125.9, 122.0, 121.3, 29.6. HRMS-ESI [M + Na]^+^ calcd for C_15_H_11_NNaO_2_^+^, 260.0682; found, 260.0686.

*2-(2-Methoxy-2-oxoethyl)phenyl 3-phenyl-2H-azirine-2-carboxylate* (**14b**). Compound **14b** was prepared following GP-H procedure from 5-chloro-3-phenylisoxazole **2a** (359 mg, 2 mmol, 1 equiv.) with methyl 2-(2-hydroxyphenyl)acetate **12b** (499 mg, 3 mmol, 1.5 equiv.) as a nucleophile and *N*,*N*-dimethylpyridin-4-amine (733 mg, 6 mmol, 3 equiv.) as a base. A mixture of PE–EtOAc (from 20:1 to 5:1) was used as an eluent for chromatography. Colorless solid (835 mg, yield 92%). Mp: 52–53 °C (Et_2_O–hexane). ^1^H NMR (400 MHz, CDCl_3_) *δ* 8.08–7.99 (m, 2H), 7.74–7.61 (m, 3H), 7.35–7.29 (m, 2H), 7.25–7.15 (m, 2H), 3.64 (s, 3H), 3.58 (AB-q, *J* = 15.6 Hz, 2H), 3.09 (s, 1H). ^13^C{1H} NMR (100 MHz, CDCl_3_) *δ* 170.9, 169.8, 158.1, 148.9, 134.1, 131.3, 130.6, 129.4, 128.5, 126.3, 126.2, 122.4, 122.0, 52.0, 36.1, 29.5. HRMS-ESI [M + H]^+^ calcd for C_18_H_16_NO_4_^+^, 310.1074; found, 310.1077.

*Prop-2-yn-1-yl 3-phenyl-2H-azirine-2-carboxylate* (**14c**). Compound **14c** was prepared following GP-H p*rocedure* from 5-chloro-3-phenylisoxazole **2a** (359 mg, 2 mmol, 1 equiv.) with prop-2-yn-1-ol **12c** (168 mg, 3 mmol, 1.5 equiv.) as a nucleophile and *N*,*N*-dimethylpyridin-4-amine (733 mg, 6 mmol, 3 equiv.) as a base. A mixture of PE–EtOAc (from 20:1 to 5:1) was used as an eluent for chromatography. Colorless solid (307 mg, yield 77%). Mp: 42–43 °C (Et_2_O–hexane). ^1^H NMR (400 MHz, CDCl_3_) *δ* 7.93–7.87 (m, 2H), 7.71–7.64 (m, 1H), 7.63–7.56 (m, 2H), 4.77 (dd, *J* = 2.4, 1.0 Hz, 2H), 2.91 (s, 1H), 2.50 (t, *J* = 2.4 Hz, 1H). ^13^C{1H} NMR (100 MHz, CDCl_3_) *δ* 170.9, 158.0, 134.0, 130.5, 129.3, 121.9, 77.2, 75.2, 52.6, 29.3. HRMS-ESI [M + Na]^+^ calcd for C_12_H_9_NNaO_2_^+^, 222.0526; found, 222.0527.

*Benzyl 3-phenyl-2H-azirine-2-carboxylate* (**14d**) [26]. Compound **14d** was prepared following GP-H procedure from 5-chloro-3-phenylisoxazole **2a** (359 mg, 2 mmol, 1 equiv.) with phenylmethanol **12d** (324 mg, 3 mmol, 1.5 equiv.) as a nucleophile and *N*,*N*-dimethylpyridin-4-amine (733 mg, 6 mmol, 3 equiv.) as a base. A mixture of PE–EtOAc (from 20:1 to 5:1) was used as an eluent for chromatography. Colorless solid (636 mg, yield 79%). Mp: 65–66 °C (Et_2_O–hexane). ^1^H NMR (400 MHz, CDCl_3_) *δ* 7.92–7.88 (m, 2H), 7.69–7.64 (m, 1H), 7.62–7.57 (m, 2H), 7.39–7.34 (m, 5H), 5.23 (AB-q, *J* = 12.4 Hz, 2H), 2.93 (s, 1H). ^13^C{1H} NMR (100 MHz, CDCl_3_) *δ* 171.5, 158.4, 135.6, 133.9, 130.4, 129.3, 128.5, 128.2, 128.1, 122.2, 66.9, 29.6. HRMS-ESI [M + Na]^+^ calcd for C_16_H_13_NNaO_2_^+^, 274.0839; found, 274.0841.

*S-Phenyl 3-phenyl-2H-azirine-2-carbothioate* (**15a**)**.** Compound **15a** was prepared following GP-H procedure from 5-chloro-3-phenylisoxazole **2a** (359 mg, 2 mmol, 1 equiv.) with benzenethiol **13a** (331 mg, 3 mmol, 1.5 equiv.) as a nucleophile and *N*,*N*-dimethylpyridin-4-amine (733 mg, 6 mmol, 3 equiv.) as a base. A mixture of PE–EtOAc (from 20:1 to 5:1) was used as an eluent for chromatography. Colorless solid (410 mg, yield 81%). Mp: 54–55 °C (Et_2_O–hexane). ^1^H NMR (400 MHz, CDCl_3_) *δ* 8.01–7.96 (m, 2H), 7.74–7.69 (m, 1H), 7.67–7.61 (m, 2H), 7.45–7.39 (m, 5H), 3.22 (s, 1H). ^13^C{1H} NMR (100 MHz, CDCl_3_) *δ* 196.8, 158.8, 134.6, 134.3, 130.7, 129.5, 129.4, 129.1, 126.9, 121.7, 38.0. HRMS-ESI [M + Na]^+^ calcd for C_15_H_11_NNaOS^+^, 276.0454; found, 276.0456.

*S-Cyclopentyl 3-phenyl-2H-azirine-2-carbothioate* (**15b**). Compound **15b** was prepared following GP-H procedure from 5-chloro-3-phenylisoxazole **2a** (359 mg, 2 mmol, 1 equiv.) with cyclopentanethiol **13b** (306 mg, 3 mmol, 1.5 equiv.) as a nucleophile and *N*,*N*-dimethylpyridin-4-amine (733 mg, 6 mmol, 3 equiv.) as a base. A mixture of PE–EtOAc (from 20:1 to 5:1) was used as an eluent for chromatography. Yellow oil (358 mg, yield 73%). ^1^H NMR (400 MHz, CDCl_3_) *δ* 7.97–7.86 (m, 2H), 7.70–7.57 (m, 3H), 3.83–3.71 (m, 1H), 3.11 (s, 1H), 2.19–2.01 (m, 2H), 1.70–1.45 (m, 6H). ^13^C{1H} NMR (100 MHz, CDCl_3_) *δ* 199.3, 158.7, 134.0, 130.6, 129.4, 121.9, 42.4, 38.1, 33.3, 33.1, 24.7, 24.7. HRMS-ESI [M + H]^+^ calcd for C_14_H_16_NOS^+^, 246.0947; found, 246.0951.

*S-(Benzamidomethyl) 3-phenyl-2H-azirine-2-carbothioate* (**15c**). Compound **15c** was prepared following GP-H procedure from 5-chloro-3-phenylisoxazole **2a** (359 mg, 2 mmol, 1 equiv.) with N-(mercaptomethyl)benzamide **13c** (502 mg, 3 mmol, 1.5 equiv.) as a nucleophile and *N*,*N*-dimethylpyridin-4-amine (733 mg, 6 mmol, 3 equiv.) as a base. A mixture of PE–EtOAc (from 20:1 to 5:1) was used as an eluent for chromatography. Light yellow solid (130 mg, yield 21%). Mp: 112–113 °C (Et_2_O–hexane). ^1^H NMR (400 MHz, CDCl_3_) *δ* 8.15 (s, 1H), 7.95–7.88 (m, 2H), 7.75–7.86 (m, 1H), 7.65–7.58 (m, 2H), 7.51–7.44 (m, 2H), 7.37–7.30 (m, 2H), 7.18–7.08 (m, 1H), 3.70 (s, 2H), 3.24 (s, 1H). ^13^C{1H} NMR (100 MHz, CDCl_3_) *δ* 199.8, 166.0, 158.2, 137.5, 134.6, 130.8, 129.6, 128.9, 124.5, 121.1, 119.9, 37.9, 33.6. HRMS-ESI [M + H]^+^ calcd for C_17_H_14_N_2_OS^+^, 311.0849; found, 311.0852.

## 4. Conclusions

Mild and rapid procedures for the preparation of amides, anhydrides, esters, and thioesters of 2*H*-azirine-2-carboxylic acids involving FeCl_2_-catalyzed isomerization of 5-chloroisoxazoles to 2*H*-azirine-2-carbonyl chlorides, followed by reaction with N-, O-, or S-nucleophiles mediated by an *ortho*-substituted pyridine have been developed. In the case of readily available chloroisoxazoles and nucleophiles, inexpensive 2-picoline is a suitable base. To achieve the maximum yield of the acylation product with 2*H*-azirine-2-carbonyl chlorides, it is recommended to use the reagent 2-(trimethylsilyl)pyridine/ethyl chloroformate, which often made it possible to increase the yield of acylation products by 1.5–3 times. The mechanism of acylation with 2*H*-azirine-2-carbonyl chlorides was discussed.

## Data Availability

Data are contained within the article.

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
