# Peer review of "5-Chloroisoxazoles: A Versatile Starting Material for the Preparation of Amides, Anhydrides, Esters, and Thioesters of 2H-Azirine-2-carboxylic Acids"

_molecules, 2022, doi:10.3390/molecules28010275_

Round 1

Reviewer 1 Report

This manuscript by Khlebnikov and co-workers describes investigation of acylation of 2H-azirine-2-carbonyl chlorides resulted from the FeCl2-catalyzed isomerization of 5-chloroisoxazoles with N-, O-, or S-nucleophiles to offer the corresponding amides, anhydrides, esters, and thioesters. Under the optimal conditions, the yields were good with the assistance of ortho-substituted pyridines. The results appear to be useful in practical preparation 2H-azirine-2-carbonyl acid derivatives. I recommend a publication after minor revisions as following.

1) The abstract is almost the same as conclusions, please improve.

2) Please provide all H & C NMR spectra in the supplementary, to clearly indicate the purity of the compounds reported.

Author Response

We are grateful to the editor and reviewers for carefully reviewing the manuscript and comments, the consideration of which made it possible to significantly improve the manuscript.

Reviewer 1

1) The abstract is almost the same as conclusions, please improve.

This has been done.

2) Please provide all H & C NMR spectra in the supplementary, to clearly indicate the purity of the compounds reported.

This has already been done.

Reviewer 2 Report

Comments

The article entitled “5-Chloroisoxazoles: A Versatile Starting Material for the Preparation of Amides, Anhydrides, Esters and Thioesters of 2H-Azirine-2-carboxylic Acids” describes the synthesis of amides, anhydrides, esters, and thioesters from 2H-azirine-2-carboxylic acids at room temperature. The process involves the isomerization of 5-chloroisox azoles to 2H-azirine-2-carbonyl chlorides by FeCl2 followed by reaction with N-, O-, or S-nucleophiles mediated by an ortho-substituted pyridine in presence of inexpensive base like 2-picoline.

However, there are some suggestions requested to the author to be included in the manuscript. 

1. If possible, please include the mechanism for the conversion of 2a to 1a, so that readers will be benefited or get better understanding from mechanistic point of view.

Author Response

We are grateful to the editor and reviewers for carefully reviewing the manuscript and comments, the consideration of which made it possible to significantly improve the manuscript.

Reviewer 2

  1. 1. If possible, please include the mechanism for the conversion of 2a to 1a, so that readers will be benefited or get better understanding from mechanistic point of view.

The mechanism of isomerization was considered using DFT calculations in [12]. The corresponding link has been added to the text.

Reviewer 3 Report

General remarks:

The work presented by Prof. Khlenikov et al. describes the preparation of amides, anhydrides, esters, and thioesters derived 2H-azirine-2-carboxylic acids through FeCl2-promoted ring-contraction of 5-chloroisoxazoles and subsequent reaction with N-, O-, or S-nucleophiles. The isomerization of 5-chloroisoxazoles has been already used by this group for the preparation of 1H-pyrroles. However, the work is quite interesting since this approach provides access to functionalized 2H-azirines, which can be important intermediates in the preparation of other compounds.

In my opinion, this new account fulfill the requirements for publication in this journal. I consider it is appropriate for the journal’s readership and I recommend publication in Molecules after a few minor remarks to take in account:

- Page 2, Scheme 1: eq. 1: ref. 4 is not correct. Change by ref. 9.

- Page 2, Scheme 1: eq. 2: I think the correct references are from 10 to 17

- Page 2, Scheme 1: eq. 3: change ref. 6 by ref. 12.

- Page 2, Scheme 1: eq. 4: change ref. 10a by ref. 19, and ref. 10b,c by ref. 20,21.

- Table 1, entry 8: I suppose you use iPr2EtN and not iPrEtNH. Anyway, the formula should be revised.

- Scheme 3: benzylamine 6g afford compound 7g, and tert-butylamine 6h give product 7h. Change the order of both compounds in Scheme 3.

- Sentence from lines 94–98 has no sense. It seems that the verb is missing.

- Page 9, lines 203, 210, 220, 231: compound numbers in boldface.

- Page 10, lines 243, 251, 275: compound numbers in boldface.

- Page 12, line 345: the integration of signal at 3.04 ppm corresponds to one proton and not three protons.

- Page 12, line 349: Triphenyl (in capital letter).

- Page 12, line 358: Diphenyl (in capital letter).

- Page 12, line 364: the integration of signal at 7.13-7.07 ppm corresponds to one proton and not two protons.

- Compound 7g in not a new compound since it has been previously prepared (ref. 16 and 17). It should be pointed out in the experimental section.

- Page 12, line 386: tert-Butyl (in capital letter).

- For compounds 7i-q the amine 2-methylpropan-2-amine or tert-butylamine 6h has been used as starting material. Change 6g by 6h in the experimental of these compounds.

- Page 14, line 454: Butyl (in capital letter).

- Page 14, line 492: for signal 3.09, (s, 1H) is missing.

- Compound 14d in not a new compound since it has been previously prepared (Chem. Sci., 2020, 11, 947–953). It should be pointed out in the experimental section.

- Page 15, line 522: compound name in italic.

- Page 15, line 531: Benzamidomethyl (in capital letter).

- Ref. 2: Chem. Rev. in italic.

- Ref. 4: Dysidea fragilis in italic.

- Ref. 5: Dysidea fragilis in italic.

- Ref. 6: Siliquariaspongia in italic.

- Ref. 8: Check journal name.

- Ref. 14: volume 49.

- Ref. 15 does not deal with metal-catalyzed isomerization of 5-amino substituted isoxazoles.

- Ref. 22: Nigst, T.A. and not Nigst, N.A.

Author Response

We are grateful to the editor and reviewers for carefully reviewing the manuscript and comments, the consideration of which made it possible to significantly improve the manuscript.

Reviewer 3

- Page 2, Scheme 1: eq. 1: ref. 4 is not correct. Change by ref. 9.

This has been corrected.

- Page 2, Scheme 1: eq. 2: I think the correct references are from 10 to 17.

This has been corrected.

- Page 2, Scheme 1: eq. 3: change ref. 6 by ref. 12.

This has been corrected.

- Page 2, Scheme 1: eq. 4: change ref. 10a by ref. 19, and ref. 10b,c by ref. 20,21.

This has been corrected.

- Table 1, entry 8: I suppose you use iPr2EtN and not iPrEtNH. Anyway, the formula should be revised.

This has been corrected.

- Scheme 3: benzylamine 6g afford compound 7g, and tert-butylamine 6h give product 7h. Change the order of both compounds in Scheme 3.

This has been corrected.

- Sentence from lines 94–98 has no sense. It seems that the verb is missing.

This has been corrected.

- Page 9, lines 203, 210, 220, 231: compound numbers in boldface.

This has been corrected.

- Page 10, lines 243, 251, 275: compound numbers in boldface.

This has been corrected.

- Page 12, line 345: the integration of signal at 3.04 ppm corresponds to one proton and not three protons.

This has been corrected.

- Page 12, line 349: Triphenyl (in capital letter).

This has been corrected.

- Page 12, line 358: Diphenyl (in capital letter).

This has been corrected.

- Page 12, line 364: the integration of signal at 7.13-7.07 ppm corresponds to one proton and not two protons.

This has been corrected.

- Compound 7g in not a new compound since it has been previously prepared (ref. 16 and 17). It should be pointed out in the experimental section.

The references have been added

- Page 12, line 386: tert-Butyl (in capital letter).

This has been corrected.

- For compounds 7i-q the amine 2-methylpropan-2-amine or tert-butylamine 6h has been used as starting material. Change 6g by 6h in the experimental of these compounds.

This has been corrected.

- Page 14, line 454: Butyl (in capital letter).

This has been corrected.

- Page 14, line 492: for signal 3.09, (s, 1H) is missing.

This has been corrected.

- Compound 14d in not a new compound since it has been previously prepared (Chem. Sci., 2020, 11, 947–953). It should be pointed out in the experimental section.

This has been corrected and the reference has been added.

- Page 15, line 522: compound name in italic.

This has been corrected.

- Page 15, line 531: Benzamidomethyl (in capital letter).

This has been corrected.

- Ref. 2: Chem. Rev. in italic.

This has been corrected.

- Ref. 4: Dysidea fragilis in italic.

This has been corrected.

- Ref. 5: Dysidea fragilis in italic.

This has been corrected.

- Ref. 6: Siliquariaspongia in italic.

This has been corrected.

- Ref. 8: Check journal name.

This has been corrected.

- Ref. 14: volume 49.

This has been corrected.

- Ref. 15 does not deal with metal-catalyzed isomerization of 5-amino substituted isoxazoles.

It deal with: “General procedure for the synthesis of 2H-azirines 1: Isoxazole (0.1 mmol), FeCl2 4H2O (2 mol %) and dry CH3CN (5 mL) were added to a 50 ml round-bottom flask. The mixture was stirred at room temperature and the reaction process was monitored until the total consumption of isoxazole. The reaction mixture was then evaporated in vacuum to remove the solvent. The crude reaction mixture was purified by column chromatography on silica gel (eluent, dichloromehane/ethyl acetate 2:10) to get product.”

- Ref. 22: Nigst, T.A. and not Nigst, N.A.

This has been corrected.

Reviewer 4 Report

This manuscript by Khlebnikov described FeCl2-catalyzed isomerization of 5-chloroisoxazoles to 2H-azirine-2-carbonyl chlorides, followed by the reactions with N-, O-, or S-nucleophiles. Previously, the author’s group has reported that pyrazoles, benzotriazoles, and sodium azide can be used as the nucleophiles to trap the 2H-azirine-2-carbonyl chlorides. However, amines, alcohols, and thiols, which are more common nucleophiles in organic synthesis, led to the complex mixtures. Using 3 equiv. of morpholine as the nucleophile, the corresponding amide was formed; but it readily decomposed in the presence of access amounts of morpholine. Therefore, they evaluated the effects of bases and found that an ortho-substituted pyridine was the best base to promote the reactions.       

The manuscript is well written and the supporting information is thorough and provides all the expected data for the new compounds. Thus, I recommend it publication in Molecules. However, the below listed minor aspects must be considered before final acceptance

1. It turned out that in the presence of 2 equiv. of morpholine, the amide 7a was formed in 70% yield. However, 1 equiv. of morpholine was added to the amide 7a, its complete destruction occurred within 10 min. It seems that 1 equiv. of morpholine is sufficient for the reaction with 2H-azirine-2-carbonyl chloride. Why the unreacted morpholine will not decompose 7a? Please comment it.

2. The reaction of morpholine with 7a can only occur in the 2H-azirine unit. The authors must provide some analysis on the possible side products.

3. Line 111, I don’t think that the observation of two diastereomers is an evidence to support several mechanisms for formation of amide 7. The chiral carbon center is not involved in the pathways in Scheme 5.

4. Scheme 1, references 10a, 10b,c, please check because each number refers to one citation.

5. Table 1, isolated yield of 6a, it should be 7a.

6. Table 1, entry 8, iPrEtN, number “2” is missed.

7. Please indicate the reaction c

Author Response

We are grateful to the editor and reviewers for carefully reviewing the manuscript and comments, the consideration of which made it possible to significantly improve the manuscript.

Reviewer 4

  1. It turned out that in the presence of 2 equiv. of morpholine, the amide 7a was formed in 70% yield. However, 1 equiv. of morpholine was added to the amide 7a, its complete destruction occurred within 10 min. It seems that 1 equiv. of morpholine is sufficient for the reaction with 2H-azirine-2-carbonyl chloride. Why the unreacted morpholine will not decompose 7a? Please comment it.

An equivalent of morpholine is consumed in the formation of morpholine hydrochloride, which does not form an amide with 2H-azirine-2-carbonyl chloride and does not decompose 7a.

  1. The reaction of morpholine with 7a can only occur in the 2H-azirine unit. The authors must provide some analysis on the possible side products.

We tried to do this, but a too complex mixture of by-products was formed, unsuitable for NMR analysis.

  1. Line 111, I don’t think that the observation of two diastereomers is an evidence to support several mechanisms for formation of amide 7. The chiral carbon center is not involved in the pathways in Scheme 5.

We agree. The misunderstanding occurred, apparently, because of the unsuccessful text in line 111 ... The text has been changed to avoid misinterpretation.

  1. Scheme 1, references 10a, 10b,c, please check because each number refers to one citation.

This has been corrected.

  1. Table 1, isolated yield of 6a, it should be 7a.

This has been corrected.

  1. Table 1, entry 8, iPrEtN, number “2” is missed.

This has been corrected.

  1. Please indicate the reaction c

We understood this comment as a need to clarify the reaction conditions in the Table 2. This has been done.